# Surface Hierarchy: Macroscopic and Microscopic Design Elements for Improved Sliding on Ice

**Karlis Agris Gross** [1,*], **Janis Lungevics** [2], **Ernests Jansons** [2], **Ilze Jerane** [1], **Michael J. Wood** [3] and **Anne-Marie Kietzig** [3]

1   Faculty of Materials Science and Applied Chemistry, Institute of Materials and Surface Engineering, Riga Technical University, LV-1048 Riga, Latvia; ilze.jerane@rtu.lv
2   Department of Mechanical Engineering and Mechatronics, Faculty of Mechanical Enngineering, Transport and Aeronautics, Riga Technical University, LV-1048 Riga, Latvia; janis.lungevics@rtu.l (J.L.); ernests.jansons_1@rtu.lv (E.J.)
3   Department of Chemical Engineering, McGill University, Montreal, QC H3A 0C5, Canada; michael.wood3@mail.mcgill.ca (M.J.W.); anne.kietzig@mcgill.ca (A.-M.K.)
*   Correspondence: kgross@rtu.lv

**Abstract:** Frictional interaction with a surface will depend on the features and topography within the contact zone. Describing this interaction is particularly complex when considering ice friction, which needs to look at both the macroscopic and microscopic levels. Since Leonardo da Vinci shared his findings that roughness increases friction, emphasis has been placed on measuring surface coarseness, neglecting the contact area. Here, a profilometer was used to measure the contact area at different slicing depths and identify contact points. Metal blocks were polished to a curved surface to reduce the contact area; further reduced by milling 400 μm grooves or laser-micromachining grooves with widths of 50 μm, 100 μm, and 150 μm. Sliding speed was measured on an inclined ice track. Asperities from pileup reduced sliding speed, but a smaller contact area from grooves and a curved sliding surface increased sliding speed. An analysis of sliding speed versus contact area from incremental slicing depths showed that a larger asperity contact surface pointed to faster sliding, but an increase in the polished surface area reduced sliding. As such, analysis of the surface at different length scales has revealed different design elements—asperities, grooves, curved zones—to alter the sliding speed on ice.

**Keywords:** ice friction; topography; texture; contact area

## 1. Introduction

Ice friction draws on the initial finding by Leonardo di Vinci, that friction between two surfaces varies linearly with the force applied to a surface, acting through the contact area. Recent explanations of ice friction have been interpreted through lower friction from a lubricant film [1] or by considering the thermodynamics at the sliding interface [2]. Contact area from the application of a load has only been approximated, emphasizing the need for renewed efforts to quantify the actual surface contact. The goal of this work was to evaluate the contact area at the macro-scale and the micro-scale to see the effect of sliding over ice and then propose further characterization at the nanoscale.

Few ice-friction studies have investigated the effect of the contact area. Initial experiments from 1939—in a cave dug out from ice at Jungfraujoch in the Swiss Alps—showed a slight increase in the friction coefficient with contact area, where the contact area varied from 2 to 300 mm$^2$ [3]. Experiments on a larger tribometer with a more pliable polyethylene slider (contact area: 200–1000 mm$^2$) on ice displayed a larger change in friction [4]. Recent experiments on contact area have shown that the increase in contact area of an ice-hemisphere loaded onto a quartz base correlates well with the friction force [5]. On a larger scale—such as with the skeleton—the contact area reduced by bending the runner

showed faster sliding [6]. These experiments have emphasized the importance of contact area, also recognized by Makkonen [2] and Lozowski [7], but only the macroscopic contact area was considered and not the microscopic and nanoscopic contacts.

*Contact Area and Surface Topography*

Contact mechanics introduced by Hertz [8]—some 400 years after Leonardo da Vinci—showed that the contact area increases with applied force, suggesting increased friction force from the more influential effect of asperities ploughing into the opposing surface. As a result, roughness as a statistical assessment of surface height variations has been commonly used to describe surfaces. A range of characterization methods provide different quantitative results with AFM most accurately measuring the total variation in the height, but is limited to a low vertical range and small analysis area. Profilometers (contact and optical) provide a more practical approach for larger areas. Many studies have insufficiently reported the roughness with a line scan—as opposed to a surface scan—to give a bearing ratio, the percentage of actual surface contact [9–11]. This "bearing ratio" or "bearing area"—coined by Abbot and Firestone [12]—as a means to more finely define the interaction surface between sliding bodies provides the basis for greater insight into physical surface features that support easier sliding.

The true contact area has been referred to in models of friction, but a report of this contact in ice friction studies remains to be seen. Baurle [4] referred to "contact spots", Bowden and Hughes [3] suggested "minute adhesions acting across the real area of contact", but Makkonen [2] outlined "contacts" at the interface over which a thermodynamic model was established. In the discussion, Makkonen suggested that an inadequate measure of the contact area increased the error. Contact area, contact points, and their location and distribution need further consideration. Measurement of the contact surface and the design of the contact surface are equally important.

Descriptive characterization of the surface has been more common, with reference to microsized morphologies in nature. The most functionality-inducing surfaces relevant to ice friction have been the tree frog's toe pad [13] and shark skin [14]. Focus on the contact surface shows that the arrangement and gaps between the tree frog's toe pads allow it to climb and adhere to extremely wet and slippery surfaces as interfacial water is squeezed and channeled out between the toe pads. In relation to the shark, fast swimming sharks benefit from a specific texture on their scales that suppresses drag at the interface: longitudinal microscopic grooves on small raised convex features [15,16]. Such grooves have a depth of 4–90 μm, width of 15–98 μm and spacing of 34–83 μm with a placement that may vary [15–19]. Microsized features also influence ice friction with spot contacts recently addressed for winter tires [20], but the microsized contacts for sliding on ice require more attention.

It is common practice in competitive ice sports to abrade blade and runner surfaces longitudinally before competing; such grooves can affect the sliding performance. If there are changes in the surface, what are the most decisive features that govern sliding over ice? This general enquiry is a question, lurking in the background, for athletes wanting to slide faster over ice. The technology versus skill level is an aspect always tested by all national teams at World Cup and Olympic events. The first step involves creating a surface that leads to faster sliding, but equally important is the characterization of the surface to determine how the contact surface structure influences sliding speed.

Attention must be focused on the total contact area: the macroscopic and the microscopic views with further consideration of the nanoscopic dimension. Previous work typically only reported roughness as an indicator of asperities on smaller areas—50 × 50 μm [21], 500 × 500 μm [9], 3 × 3 mm [10]; Rohm et al. [22] showed the importance of reporting roughness anisotropy on a larger 11 × 11 mm area, but restricted the scope of investigation to the less-sensitive focus-variation microscope.

The objective of this work was to characterize a slider's contact area and then assess its influence on sliding over ice. This involved a grooved block made by milling and by

laser machining. Attention will be directed to the macroscopic domain—contact provided from the overall shape including the laser-machined grooves (Figure 1, left)—and the microscopic domain, pointing to anisotropy adjacent to the laser-machined grooves (Figure 1, center). This study will step out from the small analysis area—previously accepted as the norm—to include an analysis of the total area. Surface features that will be addressed include the dome-like surface at the macroscale (Feature 1), together with grooves (Feature 2) and asperities at the microscale (Feature 3), as seen in Figure 1.

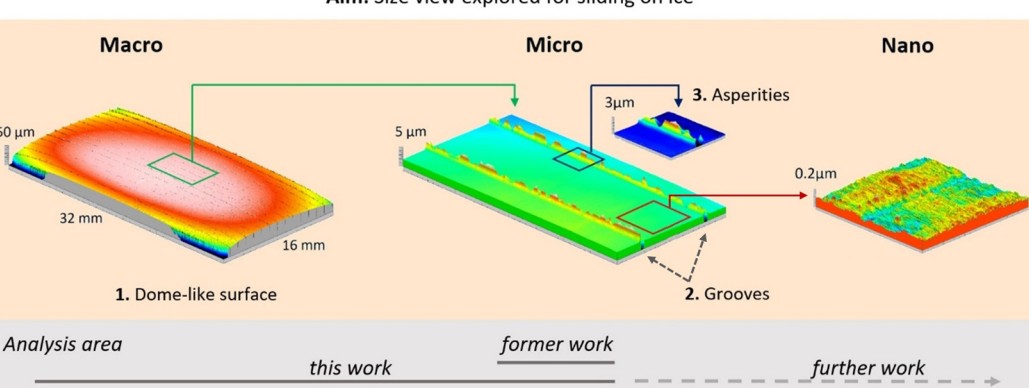

**Figure 1.** The different length scales of surface texture on the block showing the full texture, peaks/valleys, and roughness. The three features addressed in this work include (1) the dome-like surface also referred to as a curved surface; (2) grooves and the ledges that separate the flat regions; and (3) asperities on the pileup. This map of surveyable items—created from profilometer data—will be investigated in detail.

Characterization was restricted to a contact profilometer, the standard method in metrology, over the entire surface to determine the contact area at different slicing depths as a basis for seeking the influence on sliding. The hypothesis is that reduction in the contact surface from narrow grooves will lead to faster sliding.

## 2. Materials and Methods

A closer analysis of sliding on ice required a careful preparation of the sliding block surface, a detailed characterization of the sliding surface and strict control of the environmental conditions.

An array of more stringent test conditions for better experimental conditions (a) employed a test setting representative of free sliding, (b) used tight control of the environmental conditions, and (c) introduced a flattened dome-like surface on the steel block to facilitate sliding. In contrast to seeking the friction coefficient and an extrapolation to predict sliding speed, the sliding speed was directly measured for a steel block sliding down an ice-track equipped with timing sensors [23]. The variation of environmental conditions was lowered by placing an enclosure over the ice track to lower the sliding speed error. Finally, better sliding was promoted by curvature on the underside of the block, representative for many sliders in ice sports; achieved here by a more elaborate autopolishing procedure (detailed below). The collective use of these conditions facilitated easier sliding and reduced the measurement error.

### 2.1. Preparation of the Sliding Block

Ferritic stainless-steel blocks (82% Fe, 13.25% Cr, 1.75% Ni, 1.49% Mn, 0.56% Mo, 0.46% C, 0.23% V, 0.2% Al, 0.11% S) were milled to a size of $35 \times 18 \times 14 \pm 0.1$ mm and a weight of $68 \pm 0.5$ g. The blocks were initially polished before adding grooves to the underside.

### 2.1.1. Polishing for a Curved Sliding Base

The underside of the block was prepared with a raised central area by a modified polishing procedure; the intention was to prevent ploughing of the block into the ice while sliding. This central raised section was created from rocking of the block, in all directions, over the abrading/polishing surface. The slight rocking motion of the block under the push-rod was facilitated by the 0.5 mm gap in the rectangular cavity located in the support disc, Figure 2 (center, above).

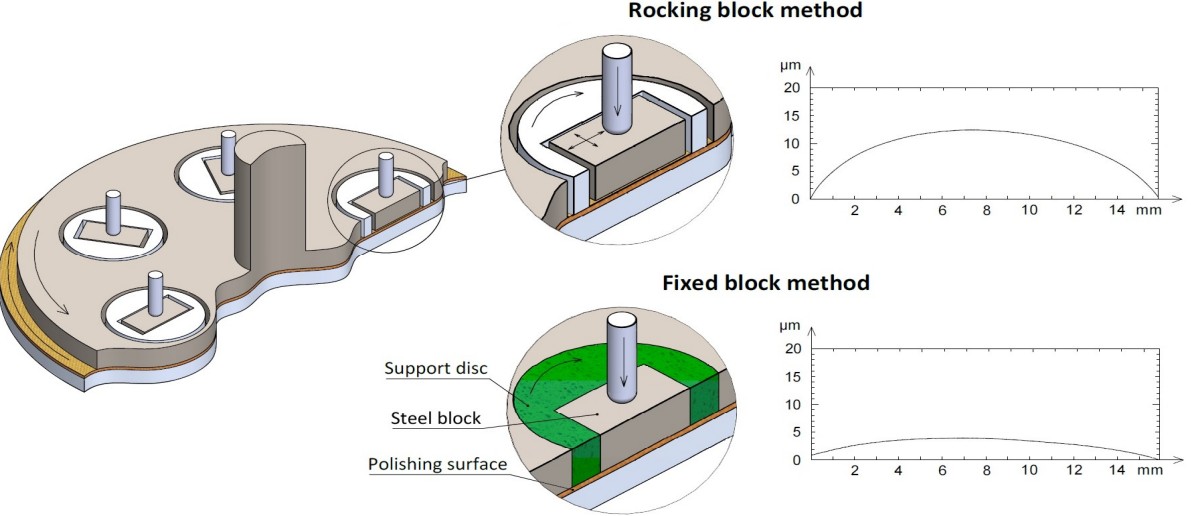

**Figure 2.** The block fixture in the moving plate of the automated polisher showing rotation from the free space around the block led to a rounded central region (top case) as opposed to a fixed block in a mount that created a flat surface.

To show that the magnitude of the central section was higher from the rocking motion during polishing, another block was fixed within the cylindrical mount and then polished—as per the right-most mounted block in Figure 2 (center, below). The fixed block was subjected to the same polishing procedure and then broken-out of the mount. The surface profile across the width through the center of the block showed that rocking created a 4-fold higher central section, Figure 2 (right, top).

Automatic polishing was conducted on an abrasive hard surface for initial leveling followed by polishing on a more elastic surface with an incremental step toward smaller diamond sizes from 9 µm, 3 µm to 1 µm; more detail is available elsewhere [24].

For a rigid surface, grooves can be extended over the entire sliding surface. This work will consider 120 µm wide milled grooves in addition to laser machined parallel grooves of three different widths: 50 µm, 100 µm, and 150 µm.

### 2.1.2. Milling and Final Polishing

Parallel grooves were milled into the polished block surface to a width of 0.4 mm, a depth of 120 µm, and 0.6 mm between groove centers. Previous work has shown that the decrease in surface area is the first means of decreasing the area upon which friction occurs, and that any depression is more favorably oriented parallel to the sliding direction [11]. Grooves were milled with a Mazak Vertical Center Smart 530C milling machine. Given the limitation in the smallest groove width by milling, narrower grooves were introduced by laser machining.

### 2.1.3. Laser Machining

A Coherent Libra Ti:sapphire femtosecond laser system (Coherent, Inc., Santa Clara, CA, USA) was used to micromachine deep grooves into mirror-polished stainless-steel blocks. This laser system has an inherent wavelength of 800 nm, pulse duration <100 fs, repetition rate of 1 kHz, and a maximal output power of 4 W. A half-wave plate and a polar-

izing beam splitter reduced the output power to the desired processing power of 200 mW. The stainless-steel substrates were mounted onto a computer-controlled 3D translation stage (Newport, Corp, Deere Avenue Irvine, CA, USA). The linear scanning velocity was set at 5 mm/s for all micromachining routines used. The laser beam was focused onto the stainless-steel substrates using a plano-convex lens (Thorlabs, Inc., Newton, NJ, USA) with a 50 mm focal length.

Three distinct inscribed geometries were designed for machining with the laser system: 50 μm wide × 50 μm deep grooves, 100 μm wide × 100 μm deep grooves, and 150 μm wide × 150 μm deep grooves, spanning the entire length of the polished stainless-steel block. In all cases, the edge-to-edge distance between consecutive grooves was held constant at 1 mm. The pristine polished stainless-steel blocks were ultrasonically cleaned in acetone for 15 min prior to irradiation and again after micromachining for 15 min to remove any non-sintered nanoparticles.

### 2.2. Measurement of Texture

The surface texture was measured by a contact stylus profilometer (Taylor Hobson Form Talysurf Intra 50) with a 2 μm tip (112/2009 stylus). An area of 32 × 16 mm was measured, representing 81% of the surface. The very edge was not included in the measured zone to prevent exceeding the limit of measurement in the z-direction, leaving the outer 1 mm unmeasured around the perimeter of the block. Texture was measured from 600 profiles: each line scan contained 6000 points to give 3,600,000 points with a point density of 7031 points/mm$^2$. Data processing was made with the Talymap Gold analysis software and initially involved levelling. Roughness (Sa), skewness (Ssk), kurtosis (Sku), peak density (Spd), and peak curvative (Spc) were calculated on the pile-up and polished sections.

The contact surface was determined by slicing at a predetermined distance from the highest asperity—digitally extracting data from the outermost layer. Slicing the top 1 μm, 2 μm, and 4 μm layers allowed visualization of the contact areas at different depths; the contact surfaces at each depth were associated with contour maps to show the microscopic contact region size and location (Figure 3). Detail on the microscopic contact area was quantified at submicron depth increments from the top-most surface to obtain a contact area profile versus the slicing depth requiring slicing up to 18 μm from the top of the surface. Slicing together with detailed reporting enabled microscopic reporting of the contact area in contrast to the previously accepted macroscopic report from the bearing ratio (compared in Figure 3).

Contact area at a given slicing depth collected incremental contributions up to the stated depth—taking input from asperities with vertical/angled walls and more gradual changes. Contact area at the slicing depth was expected to provide the best view (Figure 3).

The zoom function magnified the view of selected locations. An area of 4 × 2 mm was selected to view the smoothness of the polished area (nanoscopic view), and pileups on the edges of laser-machined grooves (microscopic view). Roughness of the polished area between the grooves (2 × 1 mm) was calculated according to ISO 25178 [25] after removing the form, the waviness, and then using a cut-off of 0.08 mm for the measurement. The profile extraction function was applied to obtain additional surface views; for example, determining the shape of the corners.

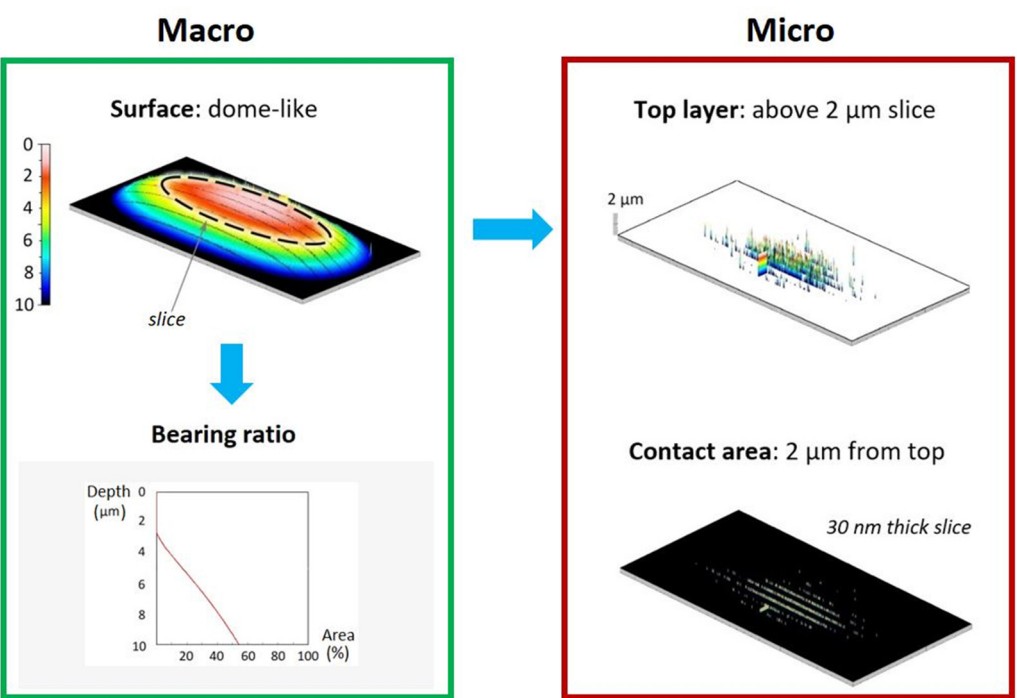

**Figure 3.** "Slicing depth" at 2 μm from the surface (**left**) shows the microscopic view of the contact area when the object sinks 2 μm deep into the ice surface (**right**). This contact area from digitally "slicing" (**right**) is contrasted with the 2 μm top layer (**right**) that shows incremental features in the surface.

*2.3. Sliding Speed*

Two types of grooved blocks were tested for sliding speed: blocks grooved by milling and by laser machining. The milled blocks were slid 20 times on ice after milling, repolished, and slid another 20 times to determine the effect of repolishing. Laser machined blocks (with 50 μm, 100 μm, and 150 μm grooves) were slid only after laser machining and were slid 20 times.

The time for sliding blocks down a 2.74 m ice track (inclined at $16 \pm 0.5°$) was measured by timing sensors. Briefly outlining the ice preparation, tap water was frozen overnight in the rectangular section, levelled by planing, and made smooth by applying a thin layer of warm water, thereby creating a pore-free top layer. Reference blocks were slid on the ice track as controls.

The temperature and humidity were maintained constant in an enclosure placed around the ice track to further reduce any departure from the conditions set in the climate simulation chamber. The ice temperature was measured with a thermocouple TP-122-100-MT-K (Czaki, Rybie, Poland) connected to a Proscan520 (Dostmann electronic GmbH, Wertheim-Reiholzheim Germany). More details on this approach for directly measuring the ease-of-sliding, as opposed to the ice friction, is detailed in Jansons et al. (2021) [11]. The improved stability of the air humidity, air temperature, and ice temperature are shown by smaller error bars with the enclosure (termed "closed") when compared to without the enclosure (labelled "open"), as seen in Figure 4.

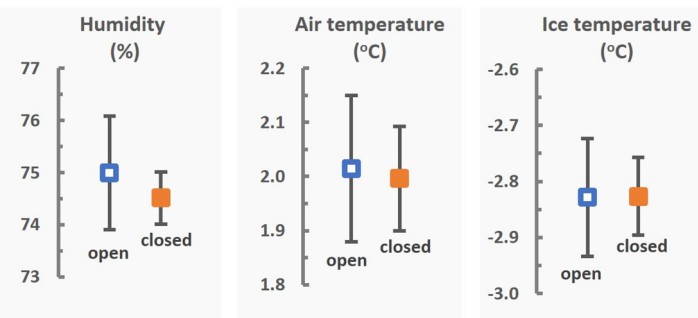

**Figure 4.** The enclosure (labelled "closed") reduced air and ice condition variability.

Two air/ice conditions were chosen to represent two different ice conditions that are the most reported and vary with respect to the water layer thicknesses on the ice. The colder ice temperature is representative of the mixed friction/boundary lubrication regime (thinner water-like layer), while the warmer ice temperature is more characteristic of hydrodynamic lubrication (thicker water-like layer), as seen in Table 1. Within this work, we did not relate the conditions to a water film thickness due to the inability to report a numerical value. Sliding speed experiments on a skeleton in a bobsled push-start facility have, however, shown a significant influence of the air humidity [26], and so humidity, air, and ice temperature were selected to match colder conditions as closely as possible in our experimental setup.

**Table 1.** Air and ice conditions for the sliding blocks (milled, laser-machined).

| Conditions | Milled Block | | Laser-Machined Blocks | |
|---|---|---|---|---|
| | Milled | Repolished | Ice1 | Ice2 |
| Air humidity | 65% | 63% | 68% | 63% |
| Air temperature | −1.0 °C | −3.0 °C | −1.2 °C | −4.3 °C |
| Ice temperature | −4.0 °C | −8.0 °C | −2.2 °C | −7.8 °C |

A correlation was sought between the sliding speed and the contact area (calculated at different slicing depths) at increasing slicing depths. The contact area was matched to the average sliding speed for the block in question, giving three data pairs that were plotted and the correlation coefficient was calculated. The correlation coefficient was determined over a range of slicing depths.

## 3. Results and Discussion

Aspects of contact area and asperities will first be addressed on a block with wider grooves, followed by a new "contact area at slice-depth" measurement method to assess contributions from asperities and a curved surface, and then we finally used this new measurement method to show the contribution of micro-sized and macro-sized contact on a laser-grooved block with narrower grooves on sliding speed.

### 3.1. Milled Surface Grooves Influence Sliding Speed

Milling of the polished block introduced 0.175 mm wide ridges, separated by 0.4 mm wide grooves. Mechanical cutting introduced an angle in the groove wall. Unintended pileups of deformed metal—resulting from cutting by the milling tool—remained on both sides of the ridge. The pileups varied in height and were unsymmetrical on each ridge, extending 8 μm to 20 μm from the top surface, as seen in Figure 5 (top, left). In some instances, even a double-peaked pileup was observed.

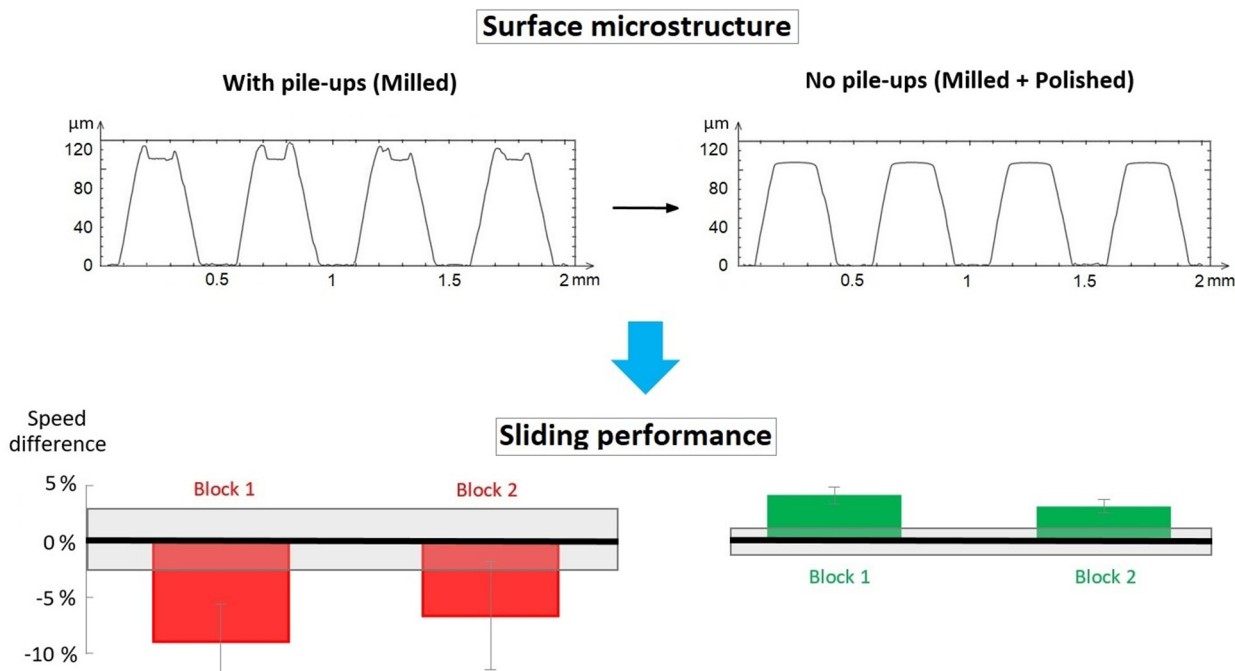

**Figure 5.** The grooved block exhibited asperities from pileups after milling (**left**) and flat ridges after repolishing (**right**). The sliding speed of blocks with pileups was slower relative to the flat surface, but faster when the pileup was removed by polishing.

Polishing the grooved block removed the pile-up, flattened, and rounded the ridges. The ridge width increased from about 0.175 mm to 0.190 mm, introducing a gentle rounding at the edge of each ridge, as seen in Figure 5 (top, right). Pileup was removed with a marginal reduction in the groove depth. Polishing-out the pileup removed the microscopic aspect and returned a nanoscopic roughness on the ridge surface to 13 nm.

Sliding speed was influenced by the reduced contact area introduced by grooving and the pileup remaining after machining. A comparison was established to an ungrooved reference block—shown as a black reference box in Figure 5. Sliding speed was slower with pileup on the grooved block when compared to the ungrooved flat block, Figure 5 (lower figure). With the pileup removed, the sliding speed was faster than the ungrooved block. This collective information shows the larger influence of surface roughness from asperities, possibly explaining the easier detection by Leonardo da Vinci. With surface roughness removed, a smaller contact area on the grooved block aided faster sliding on ice. This emphasized the influence of both microsized contacts and macroscale contact area on the sliding speed.

The pileup also increased the sliding speed error. The difference in temperature between the two testing days increased the error by 50% for the polished reference block, but the error was twofold to threefold higher for the blocks with pileup. Sliding speed for the blocks were as follows: sliding speed for milled Block 1 was $1.552 \pm 0.053$ m/s and $1.592 \pm 0.077$ m/s for milled Block 2, compared to $1.705 \pm 0.058$ m/s for the polished block to which the results were compared. An increase in sliding speed after removing ridges with polishing was clear, showing a sliding speed of $1.898 \pm 0.024$ m/s for Block 1 and $1.868 \pm 0.025$ m/s for Block 2 when compared to $1.812 \pm 0.045$ m/s for the ungrooved polished reference block.

Sliding for the milling-grooved blocks was improved by the absence of asperities and a smaller contact area imparted by grooves.

### 3.2. Characterization of Laser-Grooved Blocks

The contour maps of Figure 6 represent three laser-engraved blocks with grooves as black lines and rounded corners from the rocking-block polishing method. The size of

the central plateau from the rocking-block polishing method varied amongst the blocks (Figure 6a). The smallest plateau was found on the 50 μm laser-grooved block, as seen by the smallest encompassing ellipse that shows the 1 μm top-layer; the 100 μm wide grooved block exhibited a more elongated 1 μm top-layer, but the block with 150 μm grooves showed the largest top 1 μm contact footprint. Since larger contact areas are associated with more friction [21,27,28], then the initial intuition could be that the last block would have the slowest sliding speed. Reports on the sliding speed, however, indicated the opposite trend.

**Figure 6.** (**a**) Contour plots with a larger vertical scale show a central raised section and laser-machined grooves. The contours display the location and placement of the upper 1 μm, 2 μm, and 4 μm layers on the overall surface form. Photo-simulated corner views show rounding from the rocking-block polishing method and pileup adjacent to the grooves. (**b**) Contact area was considerably smaller than the footprint area for the 50 μm grooved surface in the 1 μm, 2 μm, and 4 μm top layers with more pronounced asperities.

The contours do not represent the contact area, but the footprint that encloses the highest points. The actual contact area at the three different depths (1 μm, 2 μm, and 4 μm) was significantly smaller than the footprint area; the contact area at a depth of 1 μm for the 50 μm, 100 μm, and 150 μm grooved blocks was 0.11, 5.77, and 29.2 μm$^2$, respectively. Contact area at 1 μm from the top surface differed by as much as about 300 times between the 50 μm grooved block and the 150 μm grooved block.

Digital slicing of the grooved blocks shows points of contact—a useful addition to contour maps. The color gradation within each slice, provided by the analysis software, aids in the interpretation of the height (Figure 6b). The small 0.11 μm$^2$ true contact area in the 1 μm slice of the 50 μm grooved block can be attributed to sparsely populated pileup within the larger footprint. Contact points on the 100 μm grooved and the 150 μm grooved

blocks were less associated with asperities, but more with the rounded surface (Figure 6a). This closer view at the microscopic level provided closer examination of the contact area than the contour plot.

A close-up examination revealed that asperities adjacent to the grooves appeared as long narrow sawtooths (Figure 1—microscopic view). Pileups from the line scan were 1.2 μm high for the 50 μm grooved surface, and between 2 μm and 3 μm for the other two laser engraved surfaces (see Supplementary Figure S1).

The photosimulated view of the corner area (at 2 mm from the edge) provided more detail on the shape and asperities (Figure 6a). Rounding at block corners created a 20 μm height difference with the center of the block (Figure 6). Such rounding from the rocking-block polishing method presents an advantage for better sliding on ice.

Contact area visualization on the total block revealed three topographical features: (1) a central plateau, (2) ridges, and (3) asperities from laser-machining. Primary attention was given to roughness, as initially pointed out by Leonardo da Vinci [29], and often shown by ice-friction researchers. Roughness along the saw-toothed microscopic pileup was largest on the 50 μm laser-grooved block with Sa at 0.58 μm (Table 2). Roughness on the surrounding polished ridges exhibited nanoscopic roughness—between 6 and 8 nm for all blocks. This information announces the possible dominating role of roughness, but a larger number of contact points per unit area and the associated contact area needs to be evaluated. The contact points (Spd) were not resolvable on the pile-up due to filters in the analysis software.

**Table 2.** Roughness characteristics of the laser-grooved and milling-grooved blocks.

| | Texture Parameters | | | | |
|---|---|---|---|---|---|
| | Sa Nm | Ssk | Sku | Spd pks/mm² | Spc 1/m |
| Laser-grooved block, along pileups | | | | | |
| 50 μm | 581.1 | 1.04 | 3.53 | - | - |
| 100 μm | 65.3 | 0.15 | 5.03 | - | - |
| 150 μm | 49.4 | 1.23 | 8.51 | - | - |
| Laser-grooved block, polished zone | | | | | |
| 50 μm | 7.7 | −0.02 | 3.09 | 7971 | 0.029 |
| 100 μm | 6.3 | −0.09 | 2.97 | 5440 | 0.026 |
| 150 μm | 6.4 | 0.02 | 2.89 | 3823 | 0.021 |
| Milling-grooved block | | | | | |
| polished | 13.4 | −0.96 | 10.7 | 20654 | 0.126 |

As expected, the SEM micrographs did not detect a curved surface, and little signs of a ridge or asperities (see Supplementary Figure S2) when compared to the information generated from profilometry. It is noteworthy that laser-machined grooves show distinct, and for laser machining at the respective fluence, characteristic hole structures. These microfeatures are typically superhydrophobic [28] and can serve as traps for wear debris [30,31].

- Sa—the arithmetic mean height of asperities.
- Ssk—skewness to indicate peaks or holes above the mean plane.
- Sku—kurtosis to represent the sharpness of peaks.
- Spd—density of peas per unit area.
- Spc—the arithmetic mean peak curvature shows whether the asperity is pointed or rounded.

Grooving was conducted on stainless-steel blocks (82.00% Fe, 13.25% Cr, 1.75% Ni, 1.49% Mn, 0.56% Mo, 0.46% C, 0.23% V, 0.20% Al, 0.11% S).

### 3.3. Quantification of Contact Area

The expected contact area was determined from the analysis of contacts on the sliding surface with reference to hierarchical features—asperities, grooves, and the curved surface. Contact area at different slice depths parallel to the surface at submicron increments first showed the initial contact made with asperities, followed by further contact from the curved block surface. This situation is based on the penetration of asperities into ice and the increase in contact area arising from the larger asperity base, new asperities, and contact on the macroscopic curved surface. Further discussion will be directed to the "contact area vs. slice-depth" curve (Figure 7).

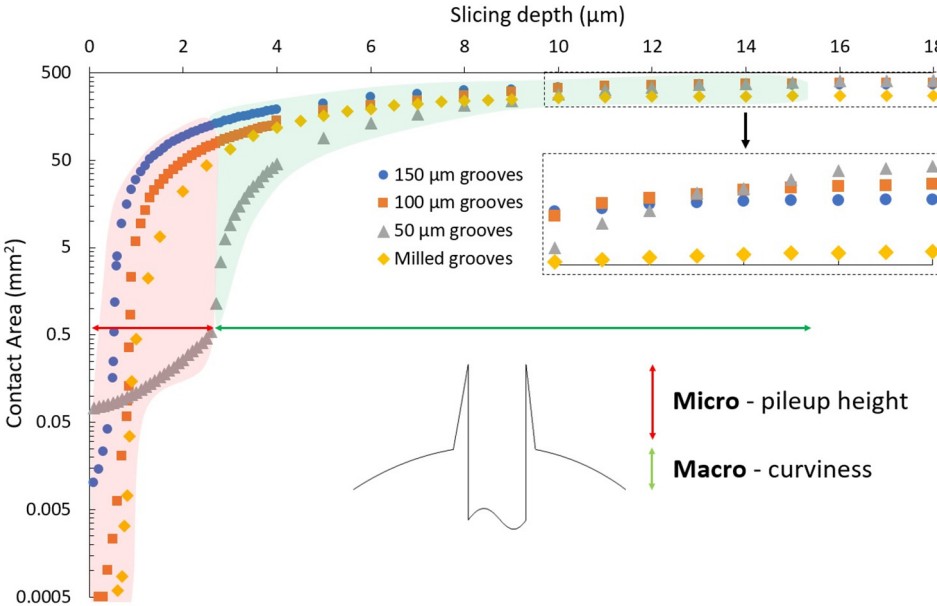

**Figure 7.** Contact area with slice depth shows minimal pileup for the 100 μm and 150 μm laser-grooved blocks, as shown by the rapidly rising contact area with slice depth, but a larger contact from numerous asperities on the 50 μm laser grooved block. The case for the 50 μm laser grooved block is accented with the red shading, showing the asperities, and the green shading shows the main contribution from the curved surface. The larger contact area for the 50 μm laser-grooved block becomes evident at a 16 μm slice depth from the highest asperity. The repolished block with milled grooves is shown for comparison.

The contact area with the slice-depth curve showed more micro-contact characteristics, an aspect that was not evident in the bearing-ratio representing the percentage of surface contact. Two aspects are noteworthy: the initial contact area and the change in contact area with slice depth. The smallest initial contact was noted for the 100 μm laser-grooved block. This suggests that few peaks, small in size, established the first contact. The same small initial contact was observed for the milled-grooved block, but from rounded peak contact (formed during the polishing process). The first contact is a feature unique to the "contact area vs. slice-depth" curve.

The change in contact area with depth was seen in more detail in the slice-depth curve as opposed to the bearing–ratio curve. A logarithmic gradation measure of contact area allowed the feature height to be determined: asperities on the 50 μm laser-grooved block were the highest amongst all blocks, as seen in the relatively constant contact area with slice-depth (shown by the red horizontal arrow in Figure 7, indicating the microscopic dimension). A similar, nearly undetectable, trend was recognized for the milled-grooved block at the initial contact, suggesting a very small asperity height. In Figure 7, this micro-hierarchical feature for the 50 μm grooved surface—highlighted in the red-shaded region—allows for the estimation of the asperity height. More detailed information of the microscopic contact builds on the feature height seen as the initial step in the bearing–ratio

curve, as confirmed by the more-constant initial contact area for the taller asperities on the 50 μm grooved surface (Figure 8).

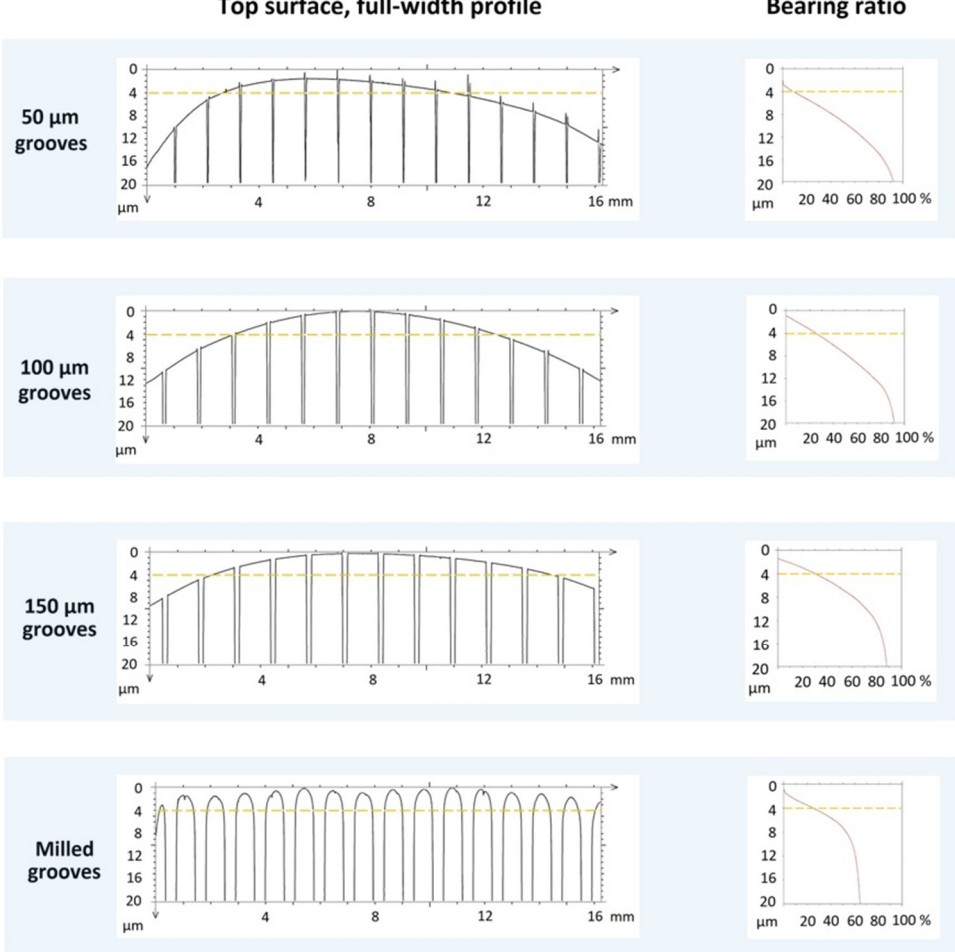

**Figure 8.** The profile of grooved blocks at a vertical magnification shows the straight edges after laser-grooving (50 μm, 100 μm, and 150 μm grooves) and the rounded edges on the milled and polished blocks. The bearing ratio curve shows the asperity height at the start with the initial vertical displacement, indicating a larger step with the 50 μm grooved block.

The added benefit of the slice–depth curve (Figure 7) is the potential to measure the steepness of the microsized features. Sloped asperities in the case of the 50 μm grooved block appear as a sloped line in the "contact area versus slice depth" curve representing the micro-zone (see Figure 7); asperities with a vertical wall would appear as a horizontal flat line, but less-steep asperities would exhibit a more rapidly rising line in the micro-zone. This explanation holds true for simpler situations for asperities in the top layer followed by a flat or curved base underneath. Therefore, the "contact area with slice-depth" curve holds more information than the conventionally used texture parameter Sdq (root mean square gradient). A complete picture of the viewed area is recommended to confirm the interpretation of the asperity shape. Herein lies an opportunity for more detailed characterization of asperity size and shape to more accurately interpret the asperities that interact with opposing surfaces.

Slicing at different depths allows the texture to be evaluated in a subsurface layer to observe the finer asperities and larger more continuous surface contribution to the contact area. This analysis could prove useful to predict the interaction of the surface under heavier loading conditions, where the surface further sinks into and further interacts with the opposing surface. For ice, this increasing contact area situation will arise at warmer ice

conditions where the slicing surface can slide into the softer ice surface. The additional interaction could then be individually assessed. This supplementary contact area from a deeper layer could be visually assessed after compiling the contact area from the different slices. Along with the need for a more detailed examination of texture within each slice will be the need for higher resolution measurement from profilometer systems: this greater resolution may result from combining results from different measurement systems such as the profilometer with atomic force microscopy data, or the continual development of profilometer measurement systems.

### 3.4. Sliding Speed of Laser-Grooved Blocks

Repeatability in the sliding speed was improved from a number of factors. The error in the sliding speed was significantly reduced with the enclosure placed around the ice track (giving as much as a three-fold smaller error), increasing the sensitivity to discern difference (Table 3). On Ice1 at $-2.2$ °C, the sliding speed of the 50 µm grooved block was $2.101 \pm 0.007$ m/s; for the 100 µm grooved block $2.139 \pm 0.015$ m/s; for the 150 µm grooved block $2.187 \pm 0.008$ m/s; and $2.177 \pm 0.020$ for the milling-grooved reference block. On Ice2 at $-8.8$ °C, the sliding speeds were $2.159 \pm 0.012$ m/s, $2.181 \pm 0.011$ m/s, $2.220 \pm 0.005$ m/s, and $2.216 \pm 0.019$ m/s for the same blocks.

**Table 3.** Contribution of enclosure conditions, ice temperature, and block surface to the sliding speed error, with reference to the sliding speed. Grey shading denotes similar settings for an easier comparison.

| Environmental Conditions | | Sliding Block Surface | | | Sliding | Error |
|---|---|---|---|---|---|---|
| Control | $T_{ice}$ (°C) | Preparation (Groove Width) | Form (mm) | $S_a$ (mm) | Speed (m/s) | (%) |
| w/o enclosure | −4.0 | milled 400 mm | 2 | - | - | 0.96 |
| " | −8.0 | milled 400 mm, polished | 2 | 13 | 2.216 | 0.80 |
| enclosure | −2.2 | polished, laser 50 mm | 16 | 581 | 2.101 | 0.33 |
| " | −2.2 | " 100 mm | 12 | 65 | 2.139 | 0.70 |
| " | −2.2 | " 150 mm | 10 | 49 | 2.187 | 0.36 |
| " | −7.8 | " 50 mm | 16 | 581 | 2.159 | 0.55 |
| " | −7.8 | " 100 mm | 12 | 65 | 2.181 | 0.50 |
| " | −7.8 | " 150 mm | 10 | 49 | 2.220 | 0.22 |

$T_{ice}$—temperature of ice; Form—total vertical displacement of the curved surface (left visual in Figure 8 for more detail); $S_a$—average roughness over entire surface.

Smaller errors in sliding speed appeared at colder temperatures, and on blocks with less curvature on the sliding surface (Table 3). We hypothesized that a larger error arises from the larger interaction volume between the sliding surface and the ice at temperatures closer to the melting of ice; testing at warmer temperatures therefore requires even higher accuracy in the different measures to discern the most influential factor. Again, we considered that a larger curvature on a freely sliding body will add to a greater degree of freedom

and more interactions with the sides of the ice track that would increase the contact area during sliding, and hence lower the sliding speed.

Sliding speed was slowest for the 50 μm grooved block, intermediate for the 100 μm grooved block, and fastest for the 150 μm grooved surface. Reference was made to the fastest sliding milling-grooved-polished block, as reported numerically above (Table 3).

Since the milling-grooved block was made with the smallest contact area, it was expected to have a faster sliding speed. The sliding speed for the 150 μm wide laser-grooved block showed a slightly higher speed than the 400 μm wide milling-grooved-polished block, accenting the contact surface at the macroscale.

Differences in sliding speed needed consideration of the surface area (after compounded effects from "rocking-block polish" and laser-grooving) and asperities remaining from laser machining. By determining the correlation of sliding speed with contact area, it was conjectured that a high correlation would be found to indicate the depth of asperity penetration into the ice surface, the results, however, showed a positive correlation with slice depths up to about 8 μm and a negative correlation at depths greater than 14 μm (Figure 9). The positive correlation from an increased contact area suggested that sliding was dictated by an effect from asperities; a larger contact area possibly distributed the load amongst more contact points, causing less asperity penetration into ice, and less resistance to sliding. The negative correlation agreed well with expectations from previous work, in that a higher contact area increases interaction/friction between surfaces [11]. The overall effect resulted in slower sliding for the 50 μm laser-grooved block.

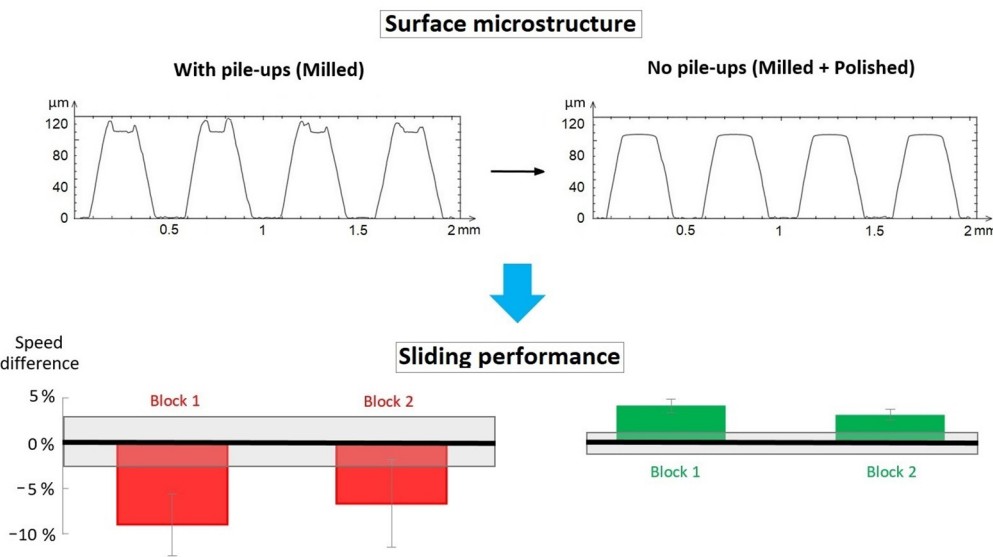

**Figure 9.** Sliding speed of laser machined blocks on ice (Ice1: −2.2 °C and Ice2: −8.8 °C) relative to grooved and polished block. The initial positive correlation suggests a faster speed from an increasing contact area from more contact point up to a slice depth of about 7 μm, followed by a negative correlation at slice depths more than 14 μm where an increase in contact area reduces sliding speed. The fastest sliding speed was 2.23 m/s.

The contact area of the 50 μm grooved surface, at slice depths showing good correlation with sliding speed—at 2 μm, 7 μm, and 15 μm—showed asperities at each slice depth (Figure 10). Such slice views are valuable to interpret the effective contact area upon penetration into ice, and discontinuous areas on the sliding surface. The resistance to sliding from asperities appeared both at the center and on the curved corners. Therefore, during stable sliding, the asperities in the center were active (see Figure 6), and if there was any rocking, then the asperities on the curved corners would also delay sliding (Figures 6 and 9).

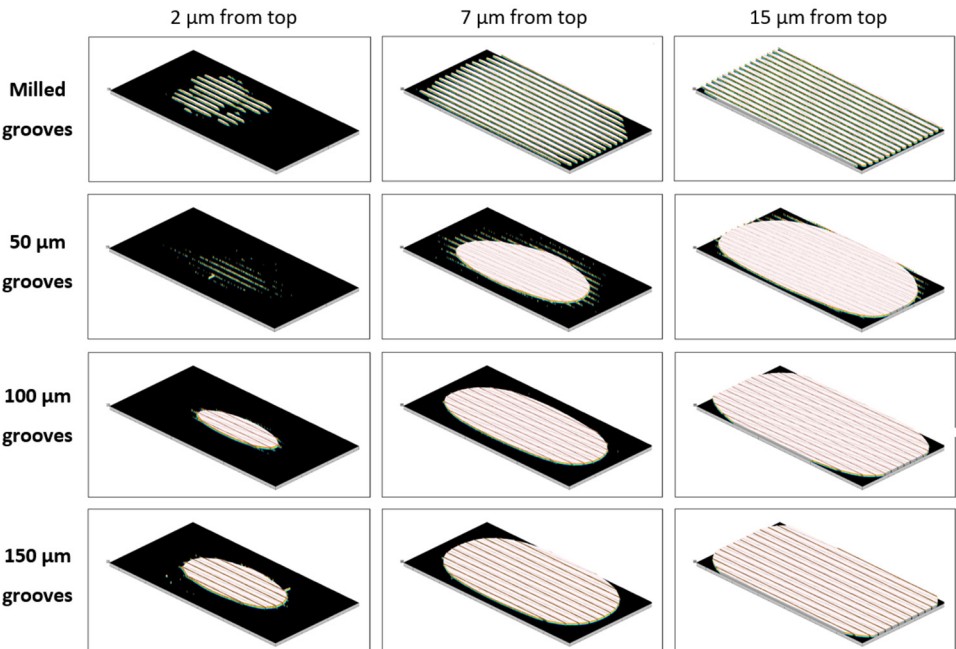

**Figure 10.** The contact area at 2 μm, 7 μm, and 15 μm slices (0.1 μm thick layer) showing the contact location on the milling-grooved block, and the laser-grooved blocks with 50 μm, 100 μm, and 150 μm grooves.

A comparison of the milling-grooved block and the faster sliding 150 μm laser-grooved block showed that the rounded macroscopic shape contact area would have been more favorable for sliding. The larger error in the sliding speed for the milling-grooved block could have arisen from the more rectangular contact area shape (Figure 10). The size, shape, and location of the contact areas will individually impact the ease of sliding—a contribution that is yet to be determined.

The effect of faster sliding by the 150 μm laser-grooved block was slightly more discernable on ice closer to the melting temperature at −2.2 °C. At −8.8 °C, the 150 μm laser-grooved slider was only slightly faster than the grooved and polished reference slider, possibly due to larger resistance-to-sliding from roughness on colder ice [32]. It is interesting that larger differences in the sliding speed were observed on warmer ice, despite overall slower sliding. The larger difference in sliding speed at warmer temperatures could arise from more interaction of asperities with softer ice—ploughing has a bigger effect on warmer (and softer) ice.

Since each block has a different curved surface, number, and size of asperities, then the penetration into ice and the resulting contact area will be different for each block. The contact area during sliding could be defined by the area required to establish an applied stress equal to ice hardness. The area at each slice will allow the effective stress to be determined. For shallow slices, the area will be smaller and the effective stress will be greater than the hardness. Slices deeper into the block will increase the area and lower the effective stress until the effective stress reaches the ice hardness. At that point—where the effective stress and the hardness are equal—no more penetration occurs into ice, thereby defining the contact area. Therefore, the determination of sliding contact area first requires information on the ice hardness, which may change due to frictional heating at the front-end block contact to the trailing edge while sliding on ice.

More detail needs to be directed to the combined view of roughness (Sa) and peak sharpness (Sku) (Table 2) to see how the peak curvature would influence the sliding speed. There is a growing need to use texture parameters, "contact area at slice depth", distribution of contact points, and mapping of the surface for showing topography to determine the source of friction.

The analysis showed the dominating role of asperities and so further work could consider how asperity nanotopography interacts with ice to influence the sliding speed. Nanoscale roughness is starting to be more actively investigated in friction studies [33]. In past studies, the interacting ice surface has been either modeled as a solid surface with mobile water molecules [34–36] or interpreted by others as a liquid—termed the "liquid-like layer" [1]. Further studies need more detail and further characterization of the sliding surface and the underlying ice surface. While previous studies have considered that water evolved from friction or from the less influential pressure melting when addressing the thermodynamics at the sliding interface [2], the extension of the model to include the influence of asperities on ice-friction would make an important contribution.

### 3.5. Further Considerations on the Contact Area

This work has presented an approach to determine the contact area over a larger area of engineering significance: the location of the contacts and the total contact area at deeper slicing depths. These two new aspects were introduced to continue from Makkonen's thermodynamic model [2], which showed greater friction from a longer ridge-like asperity. This highlights the importance for the 3-D characterization of asperities.

A further development could make a 3D model of the surface texture and model the increase in surface area under changing loading conditions. Initial studies on contact areas have been measured by conductivity on a very small scale using a contact-AFM [37]. Larger contact areas have been visualized using an optical microscope through molecules that fluoresce under load [38]—a forerunner to a contact microscope [39]. Both methods characterize at a smaller scale than the profilometry that has been used here. These concurrent activities highlight active research on more precise measurements of contact between two bodies.

Ice as the sliding surface has been assumed as stable with defined properties, but recent work has revealed changes with temperature and sliding speed. A decrease in ice hardness with temperature [2,40] lowers the resistance for penetration into the ice. Presence of asperities on a slider such as with the 50 μm laser-grooved block would make penetration into ice even more pronounced from the higher contact stress. The slider contact area with slicing depth would allow a closer insight into the possible asperity penetration depth and the resulting sliding resistance. Asperity penetration depth may factor in the increase in ice hardness with sliding speed, previously noted during slow sliding speeds [41]; the change in hardness at higher speeds remains to be asserted and quantified. Finally, mechanically disturbed ice will change to heal previously made scratches [42], and in real-life conditions, will need to consider continuous changes to the ice surface from the pounding of runners during sliding over ice [26].

Previous studies have shown that a lower bearing ratio (% of slider contact) aids in a faster sliding speed [11], but like roughness, it is a statistical measure of the surface; without information on the microtopographic features and for practical use, there is an assumption of the asperity penetration depth. The bearing ratio measures the macroscopic aspect (initially proposed by Abbot and Firestone [12]), however, the "contact area versus slice depth" curve and maps of contact area at different depths show the microscopic and macroscopic contributions. Recent experiments from our group have given insights into slicing from a polishing process [43], but polishing additionally introduces rounding of the contact area. Slicing, as shown in this work, showed that the contact area locations could aid the prediction of balanced sliding or veering to one side of the ice track. The 50 μm grooved block showed an off-center central plateau (Figures 6 and 8), which would have caused erratic sliding movement from continuous impact to one side of the ice track. Assessment of contact points, contact area, and contact location will assist in the design of experiments on sliding.

Surface design elements could be implemented on a smaller scale and extended to the nanoscale. More regular contact points have a lower contact area than a flat surface and is promising in stabilizing sliding on ice. The high correlation between sliding speed

and the pileup contact area from Figure 9 suggests that smaller dome-like features, akin to those on sharkskin scales, could distribute the applied load to lower penetration into ice, leaving the question of the ideal size and spacing between contact points. Grooving on raised sections provide considerations for designing surfaces with faster sliding on ice.

Sliding resistance in real-life sees contributions from ice-friction and physical barriers: ice friction from slider asperities penetrating ice to spot contacts that adhere opposing faces and physical barriers that impede motion from both surfaces (Figure 11). Surface characterization can further include the contact area with slice depth, and the shape of asperities from both surfaces. This necessitates the importance of characterizing surface topography at the macroscale, microscale, and nanoscale and a consideration of measurement from the outermost surface to provide the contact area. All these contributing factors will influence the initial stick-slip phenomena at the initial stage of sliding and determine the speed at which ice-friction is reduced to the smallest value [44].

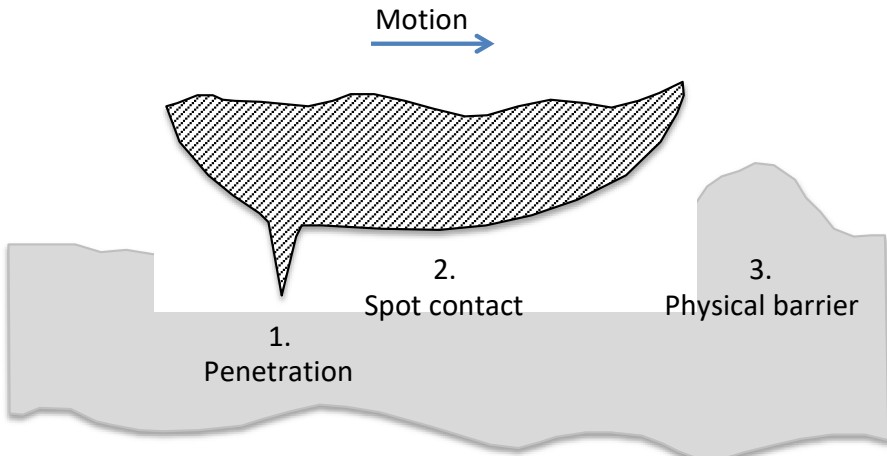

**Figure 11.** A sketch of three factors contributing to the sliding resistance: penetration of asperities, spot adhesive contacts, and physical barriers.

Further developments will benefit from providing a series of roughness parameter values, the contact area with slice depth, and a map of topographical features with more detailed characterization of the contact points. The roughness gives insight into asperity penetration into the opposing ice surface. Values of roughness (Sa), skewness (Rsk), kurtosis, and more detail on the contact point population (Ssd) and contact point geometry (Spc) will be more informative of the surface. This broadens the previous report of the surface by Scherge [10] to include information about contact points for a detailed analysis of the contact area. Measurement of these texture parameters from measurements on the more sensitive 3D laser microscope (0.2 μm probe compared with a 2 μm probe on the contact profilometer) will provide more accurate reports of the surface texture.

The hierarchical assessment of topography (specifically contact points and roughness) will be relevant to other lubricating applications such as pistons in pursuit of the lowest friction most commonly referenced to cartilage [45].

### 4. Conclusions

Sliding on ice depends on both the macroscopic and microscopic surface details of the contact area with contributions from the curved underside, grooves, and asperities over the entire surface. Microsized metal pileup decreased the sliding speed by supposedly penetrating the ice, while grooves and the domed surface jointly lowered the contact area to increase the sliding speed.

Correlation was found between the sliding speed and the pileup contact area at different slice depths, explained by more asperities lowering the contact stress and reducing

the penetration in ice. The effect of asperities was more pronounced closer to the melting point of ice, indicating deeper asperity penetration and a greater sliding contact area.

A new "rocking-block" polishing method where the block slightly rocked during polishing showed a means of introducing a curved sliding surface to lower the contact area for easier sliding over ice.

Recommendations are given to characterize the entire surface and a method suggested to measure the contact area at increasing slicing depth to provide foresight into the expected contact area at greater loading. A list of texture parameters is given to more completely describe asperities located on the sliding surface. This detailed surface characterization will find use beyond ice friction to any situation involving interactions with the surface.

**Supplementary Materials:** The following are available online at https://www.mdpi.com/article/10.3390/lubricants9100103/s1, Figure S1: An enlarged view of the grooved blocks and the associated profile for the 50 μm grooved block (upper), 100 μm grooved block (middle), and 150 μm grooved block (lower). The pileup can be seen at the edge of the grooves. Figure S2: SEM micrographs showing the grooves seen at lower magnification, the holes in the grooves at the intermediate magnification, and the pileup at higher magnification.

**Author Contributions:** Conceptualization, surface preparation—K.A.G., M.J.W., Polishing—K.A.G., Hierarchy—K.A.G. and J.L.; Investigation—E.J., I.J. and A.-M.K.; Analysis—K.A.G., E.J., I.J. and A.-M.K.; Visualization—K.A.G., J.L., E.J. and I.J.; Writing—K.A.G.; Storyline development with visuals and text—K.A.G., E.J., J.L. and I.J.; Revisions—K.A.G. and A.-M.K.; Funding acquisition—K.A.G. and A.-M.K. All authors have read and agreed to the published version of the manuscript.

**Funding:** Research was funded by the ERDF project "The Quest for Disclosing How Surface Characteristics Affects Slideability" (No.1.1.1.1/16/A/129), and a Natural Sciences and Engineering Research Council of Canada (NSERC) Discovery Grant, grant number: RGPIN-2016-04641.

**Institutional Review Board Statement:** Not applicable.

**Informed Consent Statement:** Not applicable.

**Data Availability Statement:** Data is contained within the article or Supplementary Material.

**Acknowledgments:** The authors are grateful for the assistance provided by D. Petrovics on the assembly of the enclosure, assistance in sliding experiments, and analysis of data to confirm a lower variability in air and ice conditions. Also thanks for the last reviewer, who provided valuable comments.

**Conflicts of Interest:** The authors declare no conflict of interest.

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
