# Peer review of "Surface Hierarchy: Macroscopic and Microscopic Design Elements for Improved Sliding on Ice"

_lubricants, doi:10.3390/lubricants9100103_

Round 1

Reviewer 1 Report

The manuscript submitted for review meets the requirements of the periodical.

Abstract: The object of research, the aim of the work and major conclusions have been given in the abstract. The abstract has been written in accordance with the guidelines from the periodical.

Introduction: The authors explained the need for the research in question while referring to relevant reference literature.

Materials and Methods: The chapter has been prepared in a correct way. The test method has been discussed, the research material described, and the survey instruments discussed. Minor adjustments are proposed:

  • Line 134: For better readability of the text, please present the chemical composition of the steel in the table.
  • Please improve the readability of figure 3 (chart title?).
  • Please present graphically the methodology of surface roughness measurement, laser machining and sliding speed (e.g. show the measuring stand).

Results and discussion: In the chapter containing test results, relevant graphic documentation has been presented and the results have been discussed. Minor adjustments are proposed:

  • Did the authors measure the coefficient of friction? This parameter is important in the assessment of adhesion to the surface. If so, please include the test results.
  • Figures S1 and S2 should be placed at their first mentioning in the text.

References: The reference literature consists of 44 items. The authors quote current reference literature relevant to the topic of the article. Quoting was done in accordance with the guidelines from the periodical.

Author Response

Thank you for the comments. Answers are provided in the attached document.

Reviewer 2 Report

This is a study on the effects of micro-topography on ice friction. It is of general interest, but the manuscript would be much better if the following could be considered:

  • Line 142: “ploughing” is a better terminology than “digging” in the field of tribology
  • The authors are working with the bearing curve, the Abbot curve, without mention it in that way. It would be illustrative to show this curve in lines 202-215 to better show the thinking behind “slicing”. Add a figure that describes the slicing methodology. Describe also the reference for the “top position”. This is important for the later discussion around Fig. 9.
  • There is a need to show the test setup. How does the sliding blocks look like?
  • Contact mechanics: The penetration is determined by the ice hardness since the ice is much softer than the other surface. Since the bearing ratio curves are different for different surfaces, there will be different penetration. So the same penetration cannot be compared for different surfaces. There must be a load equilibrium. Comment on how you determine the contact area at a certain load.
  • Table 2 is troublesome. The Ssk value is more or less positive for all surfaces. How come? When it is polished it must be more negative? Is “along the pileups” really along the edges of the grooves? The milled surface has clearly a negative Ssk though. The Spd parameter is also problematic. If the measurement technique uses 7031 points/mm2 then it is strange to measure 7971 peaks per mm2. Sounds as a measurement error? Or a misunderstanding.
  • Line 403: The “sliding times” must be “sliding velocities” since the unit is m/s?
  • The results in the same section, lines 397-404, are important but only mentioned in the text. It would be better to make a table with the results. Or a graph/histogram.
  • It is difficult to understand the correlation discussion. Explain what you mean by correlation.
  • Line 448: Can ploughing play a bigger role at warmer (and softer) ice? That may explain the bigger differences?

Author Response

(The authors gave the same response as above.)

Reviewer 3 Report

This is an ecxellent paper and an important contribution to understanding the effect of surface characteristics in sliding on ice. I have only some minor suggestions for improvements:

  1. The term "slicing depth" is used throughout the paper. This term is not so common, so that it might be good to explain it and perhaps even include a figure on what it means.
  2. In connection with Fig. 3, it would be good to give a small calculation on evaporation/condenstation under the experimental conditions (because condensation may affect the ice roughness). It would also be good to mention how the ice (surface!) temperature was measured.  
  3. On line 369 it reads "supposed". So, the slicing depth is based on a hypotethical penetration, which depends on the unknown hardness of ice. In Fig. 8 and the discussion below, a reader gets an impression that the slicing depth is based on the sliding experiments presented in the same figure. This should be clarified.
  4. On line 461 it is written that "nanoscale rougness is starting to be addressed in friction studies." This does not seem right, considering that already the explanation for the origin of friction is based on an analysis in the nanoscale https://doi.org/10.1063/1.3699027       
  5. Perhaps, the link of this work with the thermodynamic theories and wet friction on ice could be stengthened in the discussion. For example, ref. 2 suggests that the length (in the sliding direction) of an asperity separately affects wet friction on ice.  

Author Response

Thank you for reading the manuscript and providing comment on ways to improve the manuscript. All questions have been addressed and changes made - outlined in red- to the manuscript. Answers to the questions have been given separately.

Round 2

Reviewer 2 Report

The authors have followed my suggestions in a good way and I now suggest publication in its present form.